# Simulation of Episodic Winter Warming on Dehardening of Boreal Forest Seedlings in Northern Forest Nurseries

**Mohammed S. Lamhamedi \*, Marie-Claude Lambert and Mario Renaud**

Direction de la Recherche Forestière, Ministère des Ressources Naturelles et des Forêts 2700, Rue Einstein, Québec, QC G1P 3W8, Canada

\* Correspondence: mohammed.lamhamedi@mffp.gouv.qc.ca

**Abstract:** In recent decades, forest nurseries in eastern Canada have been faced with periods of mild winter weather, delayed snowfall, and low seedling protective snow cover combined with winter rains instead of snowfall. These extreme conditions have resulted in the loss of millions of seedlings, in particular those that overwinter outdoors, probably due to their winter dehardening. The main objective of this study is to simulate different periods of warm weather at the beginning and end of winter and evaluate their effects on the dehardening and growth of *Picea mariana* and *Picea glauca* seedlings in response to different freezing temperatures. Three warming treatments were simulated (control, 1 day, and 3 days of warming at 10 °C) followed by three freezing temperatures (−4, −12, and −20 °C). In winter, regardless of the warming treatment, the seedlings of the two species tolerated the different freezing temperatures without any apparent damage. However, at the end of winter and in the absence of snow cover, the seedlings did not show frost tolerance at −20 °C. On the other hand, the seedlings showed normal growth after undergoing frosts at −4 °C and −12 °C, similar to that observed for control seedlings. Different cultural practices and protection strategies are proposed to improve frost tolerance and reduce the winter loss of seedlings.

**Keywords:** *Picea mariana*; *Picea glauca*; forest nursery; winter warming; winter freezing; cold hardiness; growth; mineral nutrition

## 1. Introduction

Frost damage is one of the main causes of tree seedling losses in northern forest nurseries [1,2]. Due to the exposure of seedlings to harsh winter conditions in a boreal climate, winter desiccation, root frost, as well as intense early fall and late spring frosts are responsible for a considerable loss of seedlings in forest nurseries. Depending on the intensity of the frost, the estimated annual seedlings losses in each of Québec's forest nurseries vary from 5 to 30% [3]. In Ontario, Canada, evaluations conducted in the same forest nursery over several consecutive growing seasons show that the loss of seedlings caused by frost is very important (16 to 48%) [2].

During recent decades, many forest nursery managers in eastern Canada have been confronted with exceptional inter- and intra-annual variability in environmental variables characterized, for example, by daily warm temperatures above normal in both autumn and winter and late snowfall at the beginning of the winter. In addition, winter rainfall instead of snow combined with drastic changes in temperatures from warm to freezing conditions over two consecutive days and the sometimes absence of a sufficient protective snow cover are predisposing factors for the increase of seedling losses caused by winter frost in forest nurseries. For example, a severe winter frost across the province of Québec, Canada, in 2007 and 2013 caused the loss of more than 7 million and 3.9 million seedlings, respectively [4]. In some nurseries, the percentage of seedlings damaged by winter frost has exceeded 60%. Moreover, the results of a snowmaking project conducted in a forest nursery showed that the absence of a protective snow cover at the beginning of winter

alone increases the percentage of seedling mortality depending on the species and varies between 5 and 23% [4]. Recently, the diagnosis of the state of the seedlings produced in the 18 forest nurseries in the province of Québec (Canada) in 2021 revealed that 16 nurseries were affected by root frost, 17 by winter frost, 14 by spring frost, and 9 by fall frost [5]. The winter of 2021, with its climatic extremes, was the most devastating for some northern nurseries in Québec and caused, for example, the loss of nearly 11 million seedlings in a single forest nursery [6].

These losses are strongly related to different cultural practices in forest nurseries [4,7,8], such as wintering seedlings under natural conditions without resorting to the use of cold storage (−2 °C to −4 °C) to protect the seedlings against winter frost. For example, the majority of the 18 forest nurseries (6 government and 12 private) in Québec produced their seedlings in unheated tunnels during the first growing season (1 + 0). The plastic roof of tunnels was removed just before the first snow at the end of the first growing season, around mid or late October, so that the seedlings become buried under the snow during their first winter as well as their second winter before their delivery to the reforestation site [9].

However, with climate change, the frequency of these extreme events in winter will only increase in the years to come. These extreme events involve an increase in the number of days when the temperature is high and relatively long winter thaw periods. Future climate scenarios predict that temperatures will significantly increase at high latitudes during the winter compared with other seasons [10]. This could result in periods of more frequent increase in temperature and rain during winter, followed by a period of rapid freezing, which could negatively affect the survival of plant tissues (roots, stems, buds, needles, etc.), as well as the morphophysiological quality of seedlings produced in forest nurseries. Several forest nursery managers in eastern Canada (Québec) argue that a significant erratic increase in temperature during the winter for a few successive days can directly accelerate the degree of dehardening and decrease frost tolerance (also referred to as frost hardiness or cold hardiness), thereby rendering seedling tissue more vulnerable to freeze injury by subsequent frost events. Under these conditions, the ability of boreal seedlings to survive winter frosts depends not only on the levels of frost tolerance that they can reach at the end of the fall but also on their ability to remain frost-hardy during winter warm spells. However, there is evidence from forest nurseries in the southern United States that seedlings can deharden during winter in response to increasing temperature for as few as three to seven nights [11,12]. Other studies have shown that the frost hardiness of different species can be reduced or lost in a few hours to days in response to the increase in temperature [13–15]. Under such environmental conditions, if seedlings lose their hardening completely, their ability to subsequently reharden can be reduced or may be lost [15,16]. An abrupt and substantial dehardening of 3 to 14 °C has been observed in the current-year foliar of red spruce trees (*Picea rubens* Sarg.) in response to a natural thaw (3 days, 5 to 10 °C) during the winter of 1995 in northeastern North America [17].

Up until now, researchers have intensively emphasized various aspects of the hardening and dehardening of boreal tree seedlings during fall and spring [2,7,15,18]. However, despite a few studies on the effects of artificial warming in winter and spring on frost tolerance of boreal tree seedlings [19–21], the impacts of successive extreme climatic events in winter (e.g., especially a significant erratic increase in temperature followed by frost events) on the dehardening and morphophysiological quality of the whole roots and shoots of the seedlings as well as mineral nutrient concentrations, remain less investigated under forest nursery conditions. Some nursery managers believe that exposing seedlings to mild periods of winter in eastern Canada could reduce foliar nitrogen concentration. This drop in leaf nitrogen concentration could be due to a resumption of physiological activity during the winter thaw at temperatures varying between 3 and 4 °C [22]. Leaf nitrogen concentration is one of the 27 criteria for the payment of nursery managers and qualifying seedlings before they are planted in reforestation sites in Québec, Canada. For example, large seedlings (height ≥ 35 cm) must have at least a leaf nitrogen concentration of 1.6% [23].

Black spruce (*Picea mariana* [Mill.] B.S.P.) and white spruce (*Picea glauca* (Moench) Voss.) are the most widespread and used tree species in reforestation programs in North America [24]. The objectives of this study are (i) to assess the frost tolerance of one-year-old black spruce and two-year-old white spruce seedlings at the end of the fall; (ii) to simulate different periods of warm weather at the beginning and end of winter and evaluate their effects on the dehardening and growth of black spruce (1 + 0) and white spruce (2 + 0) seedlings in response to different freezing temperatures; (iii) to compare the nutritional status of black and white spruce seedlings in the fall with those that experienced mild temperatures at the beginning and end of winter; and (iv) to discuss the operational scope of these results in relation to the management of winter protection for seedlings in northern forest nurseries. Thus, to get closer to operational conditions in nurseries and reforestation activities, we used whole seedling freezing tests to quantify different freeze injuries in the entire seedling and evaluate their recovery under optimal growth conditions.

## 2. Materials and Methods

### 2.1. Plant Material

In Québec, seedlings are produced in unheated tunnels during the first growing season (May to October). Subsequently, they are transferred outside and spend the winter under the snow (January to April). Depending on the target seedling size and the volume of each cavity/container, the seedlings will be planted in a reforestation site after one year (1 + 0) in the nursery (volume of each cavity/container $\leq$ 110 cm$^3$) or kept in the nursery (volume of each cavity/container $\geq$ 200 cm$^3$) for a second growing season (2 + 0) under natural environmental conditions.

Containerized one-year-old black spruce (1 + 0) and two-year-old white spruce (2 + 0) seedlings were grown in two types of containers (67–50 and 25–310, model *IPL*, IPL inc., Saint-Damien, QC, Canada; see Table 1) at the Grandes-Piles governmental forest nursery (latitude: 46°43′54″ N; longitude: 72°42′06″ W, Québec, Canada). The seeds were sown into containers filled with a moist peat/vermiculite growing medium (3/1, *v/v*; bulk density of 0.084 g/cm$^3$). Substrate uniformity was closely monitored during potting and seeding. Once an hour (after filling 750–800 containers), a container was removed from the production line for verification of substrate density. Two weeks after germination, the seedlings were thinned to one per cavity. Seedlings were irrigated as needed to maintain the target substrate water content (45%, *v/v*) with a motorized robot (Aquaboom Harnois model, Québec, Canada) equipped with 22 nozzles and mounted on a ground rail. Water was applied at a pressure of 2.1 bars, and each pass of the robot increased the water content of the substrate by 0.9% (*v/v*). The coefficient of uniformity of this irrigation system varied between 95 and 98%. The quantities of N, phosphorous (P), and potassium (K) applied per seedling of white spruce (2 + 0) during the growing season were 261 mg, 21 mg, and 61 mg, respectively. On the other hand, the black spruce (1 + 0) seedlings received 16 mg, 5.5 mg, and 9 mg for N, P, and K, respectively. During each fertilization session, seedlings also received micronutrient elements.

In Québec, a short-day treatment is generally applied in forest nurseries towards the end of the growing season (mid-August) to improve hardening processes and frost tolerance. A black polyethylene cover, positioned approximately 40 cm above the shoot tips, was manually installed and removed each day over the seedlings to create a dark period. The short-day treatment was applied for 15 days and consisted of modifying the photoperiod of light/dark to 8 h/16 h. Other details concerning the cultural practices and production of seedlings are described in Lamhamedi et al. [25,26].

To protect the roots of the seedlings against frost at the end of autumn, the Grandes-Piles forest nursery uses a snowmaker system (model Supercrystal, Turbocristal inc., Québec, QC, Canada) to make snow (>45 m$^3$ of snow per hour) during the day and night when the air temperature is below −8 °C. The snowmaker system is distinguished by a programmable oscillation from 0° to 300°. It is manually adjustable vertically from 0° to 50° and has an application radius of 75 m when environmental conditions are ideal

(no wind, etc.). A nursery manager begins by covering the roots with a depth of 5 cm of snow. Then, they ensure total protection of the apical buds of seedlings with at least 5 cm of snow [27,28].

**Table 1.** Initial morphophysiological characteristics of one-year-old black spruce and two-year-old white spruce seedlings (mean $\pm$ SE).

|  | **Black Spruce (1 + 0)** | **White Spruce (2 + 0)** |
|---|---|---|
| Container type | Model IPL 67–50 | Model IPL 25–310 |
| Number of cavities/container | 67 | 25 |
| Volume/cavity (cm$^3$) | 50 | 310 |
| Seedling size |  |  |
| Height (cm) | 15.35 $\pm$ 0.27 | 40.25 $\pm$ 0.69 |
| Root-collar diameter (mm) | 1.65 $\pm$ 0.03 | 5.65 $\pm$ 0.12 |
| Dry mass (mg/seedling) |  |  |
| Shoot | 457.83 $\pm$ 13.47 | 8681.81 $\pm$ 359.65 |
| Roots | 253.92 $\pm$ 8.80 | 1986.79 $\pm$ 89.80 |
| Total | 711.75 $\pm$ 20.72 | 10,668.60 $\pm$ 440.57 |
| Ratio of dry mass to fresh mass (DM/FM, %) | 43.04 $\pm$ 0.49 | 46.63 $\pm$ 0.49 |
| Shoot nutrient concentrations |  |  |
| N (%) | 1.99 $\pm$ 0.05 | 2.06 $\pm$ 0.07 |
| P (%) | 0.29 $\pm$ 0.01 | 0.20 $\pm$ 0.01 |
| K (%) | 0.84 $\pm$ 0.02 | 0.59 $\pm$ 0.02 |
| Ca (%) | 0.19 $\pm$ 0.01 | 0.23 $\pm$ 0.01 |
| Mg (%) | 0.14 $\pm$ 0.00 | 0.14 $\pm$ 0.00 |

Growth variables (height and root-collar diameter) were measured at the end of the growing season on subsamples of white and black spruce seedlings (*n* = 48 seedlings per species, Table 1). These seedlings were randomly sampled from lots of seedlings in forest nurseries (500,000 to 1 million seedlings). Mineral nutrient analyses of seedlings (shoots) were determined on four composite samples (12 seedlings per composite sample per species) at the Laboratoire de chimie organique et inorganique (organic and inorganic chemistry laboratory) of the Direction de la recherche forestière (Québec Forest Research Branch, Québec, QC, Canada), using the methods described in detail in our previous studies [28–30].

The ratio of dry mass (DM) to fresh mass (FM) [(dry mass/fresh mass: DM/FM) $\times$ 100] of excised terminal shoot tips (4 cm long) was evaluated at the end of autumn (November 30) after the complete formation of buds and when the chilling sum in eastern Canada (e.g., the number of hours when temperatures < 5 °C) reached at least 200 h [3,31]. For each forest species, this ratio was determined using 4 composite samples with 12 terminal shoot tips per composite sample (Table 1). After weighing the fresh mass of the composite samples, the dry mass was determined after drying for 48 h in an oven at 60 °C.

### 2.2. Frost Tolerance of Black and White Spruce Seedlings at the End of the Fall

In order to determine the degree of hardening and frost tolerance of black (1 + 0) and white (2 + 0) spruce seedlings at the end of the fall, we evaluated various variables, including the relative electrical conductivity (RC) and an Index of Injury (It) of shoots. In addition, to verify the frost tolerance of whole seedlings (roots and shoot) at each of the freezing temperatures tested, we used a bioassay test to assess the survival and growth of the shoots and roots, as well as the bud burst of the main stem and the branches of seedlings.

2.2.1. Electrical Conductivity and Index of Injury of Excised Apical Shoot Tips in Response to Different Artificial Freezing Temperatures

The evaluation of the level of hardening of the shoots of black (1 + 0) and white (2 + 0) spruce seedlings was conducted using the electrolyte leakage method [3,32,33]. For each species, 60 apical shoot tips (4 cm long) randomly sampled from seedlings were rinsed

three times with demineralized water and placed in 125 mL Erlenmeyer flasks (3 shoot tips per flask; 20 flasks total). Then, the Erlenmeyer flasks containing the terminal shoot tips were subjected to five temperatures (T0 = 4 °C (control), T1 = −5 °C, T2 = −10 °C, T3 = −15 °C, and T4 = −20 °C). We used 4 replicates (i.e., flasks) for each temperature tested. These temperatures were simulated by a freezer (model T20RS, Tenney environmental inc., Williamsport, PA, USA) equipped with a programming controller (model Versa Tenn II, Union, NJ, USA) with a cooling rate of 2 °C/h [3,34,35].

Freezer programming sequences are described in detail in our previous studies [3,36] and other publications [37,38]. The main steps are stabilizing seedlings at a temperature of 4 °C, lowering temperatures at a rate of 2 °C/h down to 0 °C, maintaining the samples at this temperature for at least 8 h, and starting the freezing cycle at the same cooling rate. Each freezing temperature (−5 °C, −10 °C, −15 °C, and −20 °C) was maintained for one hour, after which four Erlenmeyer flasks were removed from the freezer. These samples were saturated in deionized water overnight at a temperature of 4 °C, then the electrolyte leakage (electrical conductivity measured in μS/cm) of each sample was measured (EC1, μS/cm) using a conductivity meter (model 160, Orion Research Inc., Boston, MA, USA). Subsequently, the samples were placed in an autoclave at a temperature of 121 °C for 15 min to destroy all cells and promote the maximum release of ions. After a full night at 4 °C, the leakage of electrolytes was measured again (EC2, μS/cm).

The relative electrical conductivity (RC, %) for each temperature was calculated using the following formula:

$$\text{RC (\%)} = \frac{(EC1 - B1)}{(EC2 - B2)} \times 100 \tag{1}$$

where $B1$ and $B2$ are optional blanks measured before and after oven-heating to account for possible ion leakage from the control Erlenmeyer flasks containing only deionized water [35].

For a given temperature (t), this index of injury (It) was calculated on a percentage basis by adjusting the RC of frozen samples to the unfrozen controls [39]:

$$I_t = \frac{RC_{frozen(t)} - RC_{control}}{1 - \frac{RC_{control}}{100}} \tag{2}$$

2.2.2. Growth of Seedlings, Bud Burst, and Initiation of New Roots in the Case of Whole-Seedling Freeze Treatments

For this experiment, whole seedlings were used to assess the effects of different artificial freezing temperatures (control: 4 °C, −5 °C, −10 °C, −15 °C, and −20 °C) on new root initiation, shoot growth, and bud burst. Unlike other studies that evaluate frost tolerance using the $LT_{50}$, which corresponds to the temperature at which 50% of the seedlings are considered dead [40], in this study, frost tolerance is defined as the physiological state allowing the seedling to tolerate temperatures below the freezing point without any apparent damage to the various organs of each seedling (buds, needles, branches, main stem, and roots) [3].

For each species and each temperature, 8 whole seedlings (roots and shoot) were used (40 seedlings per species total). Before the freezing tests, seedlings were placed in plastic bags (2 seedlings per bag). Once the seedlings had undergone the various artificial freezing treatments, and after overnight acclimatization at 4 °C, they were transplanted in 2 L pots filled with a standard mixture of peat and vermiculite (*v/v*: 3/1). Because of their different morphological sizes, black spruce seedlings were potted at a rate of 2 seedlings per pot, whereas for white spruce, there was only one seedling per pot. All seedlings were then placed in a greenhouse and randomly distributed in four complete random blocks (2 plants per temperature per block).

Day and night temperatures in the greenhouse were maintained at 25 °C and 18 °C (±2 °C), respectively. The photoperiod was 18 h using supplemental light provided by 400 W high pressure sodium lamps (Lumiponic Inc., Montreal, Québec, QC, Canada). Pots

were irrigated during the bioassay to maintain an optimum water content of the substrate (~45%, *v/v*) for seedling growth [27,41]. No fertilizer was applied during the experiment. After 21 days of growth in the greenhouse, the seedlings were removed from pots and the roots were washed. New white roots were separated from the initial roots of each plug. Finally, dry masses of new white roots, total shoot dry mass, and total root dry mass were determined after drying for 48 h in an oven at 60 °C.

*2.3. Simulation of Different Periods of Warm Weather and Their Effects on the Dehardening, Growth, and Mineral Nutrition of Black Spruce (1 + 0) and White Spruce (2 + 0) Seedlings at the Beginning and End of Winter*

During wintering of one-year-old black spruce (1 + 0) and two-year-old (2 + 0) white spruce seedlings under the snow in eastern Canada, Québec, seedlings of the two species were sampled at the beginning of winter (mid-January 2010) and towards the end of winter (mid-March 2010) when seedlings were not completely covered by snow. Daily maximum, mean, and minimum temperatures, and snow depth accumulation at the Grandes-Piles governmental forest nursery were estimated by linear interpolation using weather data from 8 meteorological stations near the forest nursery and the BioSIM software (version 11.8.6.2) [42] to determine the evolution of environmental variables (maximum, mean, and minimum air temperature, and snow depth accumulation) in winter during the two successive years (2009 and 2010) of this research project and the climatic normals (1991–2020).

These two successive climatic years (2009 and 2010) differed from each other in terms of snow depth accumulation and maximum, average and minimum air temperatures, as well as with the climatic normals (1991–2020) at the Grandes-Piles forest nursery (Figures 1 and 2). In fact, during this experiment, around mid-March 2010 and unlike the roots, most of the shoots of the white spruce and black spruce seedlings were not completely covered by snow (Figure 1b). This was an exceptional year as the snowmelt was early due to certain episodic rainfall.

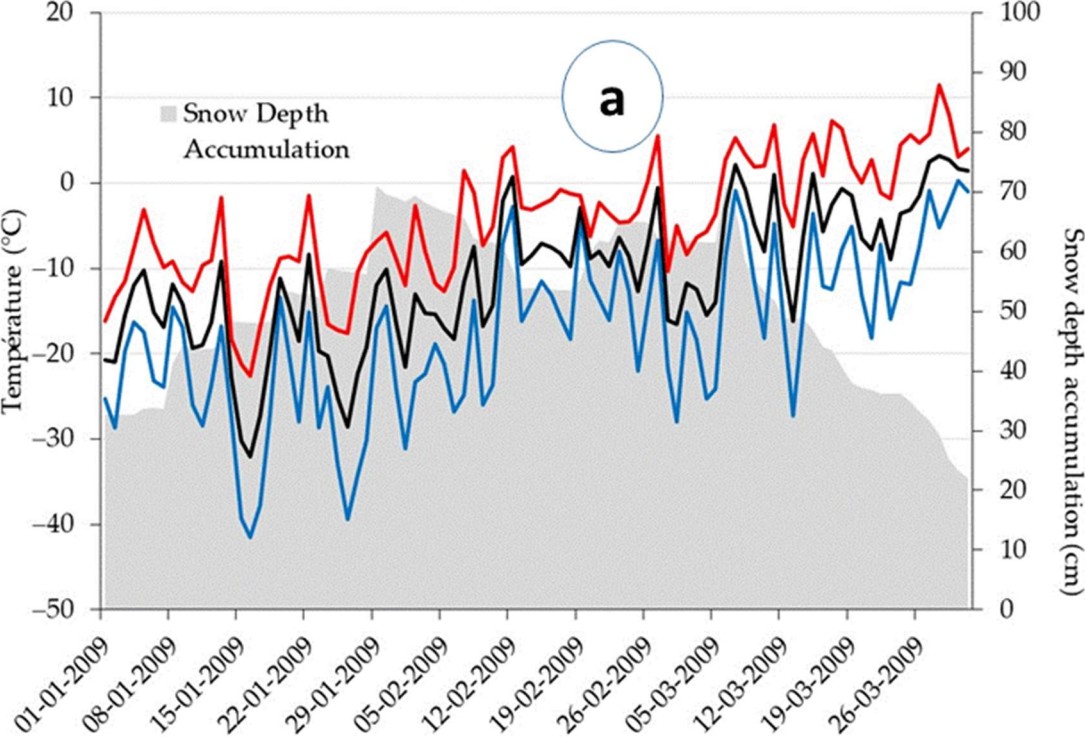

**Figure 1.** *Cont.*

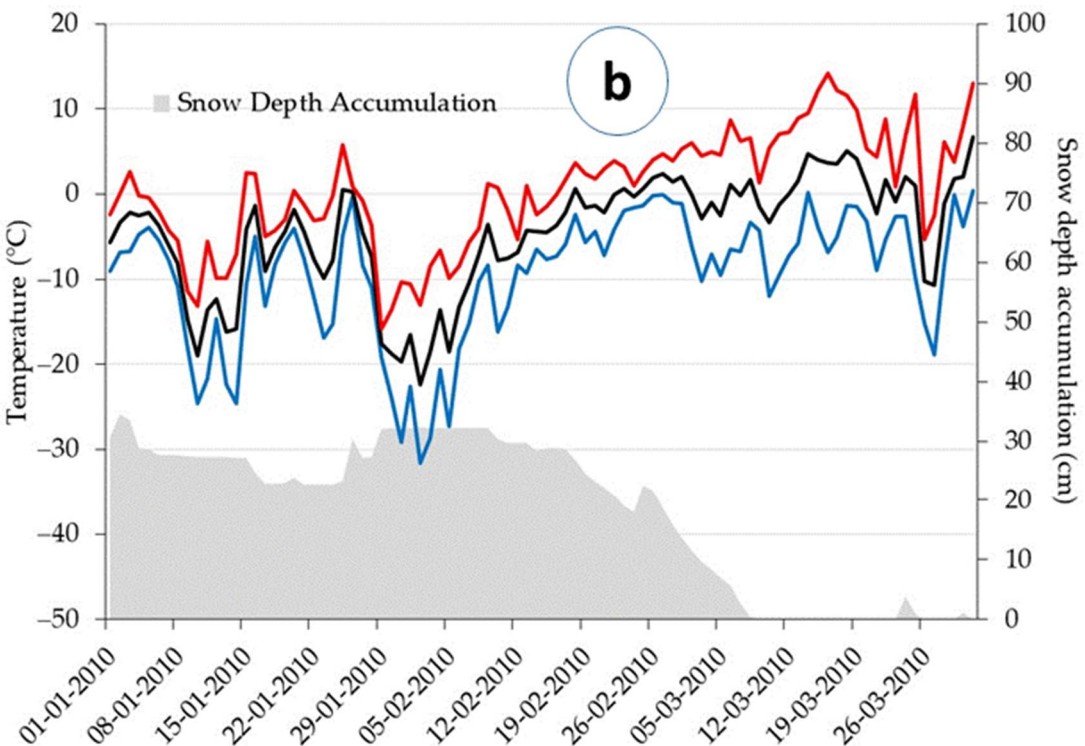

**Figure 1.** Variation of maximum (red), mean (black), and minimum (blue) daily air temperatures (January-March) at the Grandes-Piles forest nursery (Québec, QC, Canada) in (**a**) 2009 and (**b**) 2010. The areas in gray indicate snow depth accumulation.

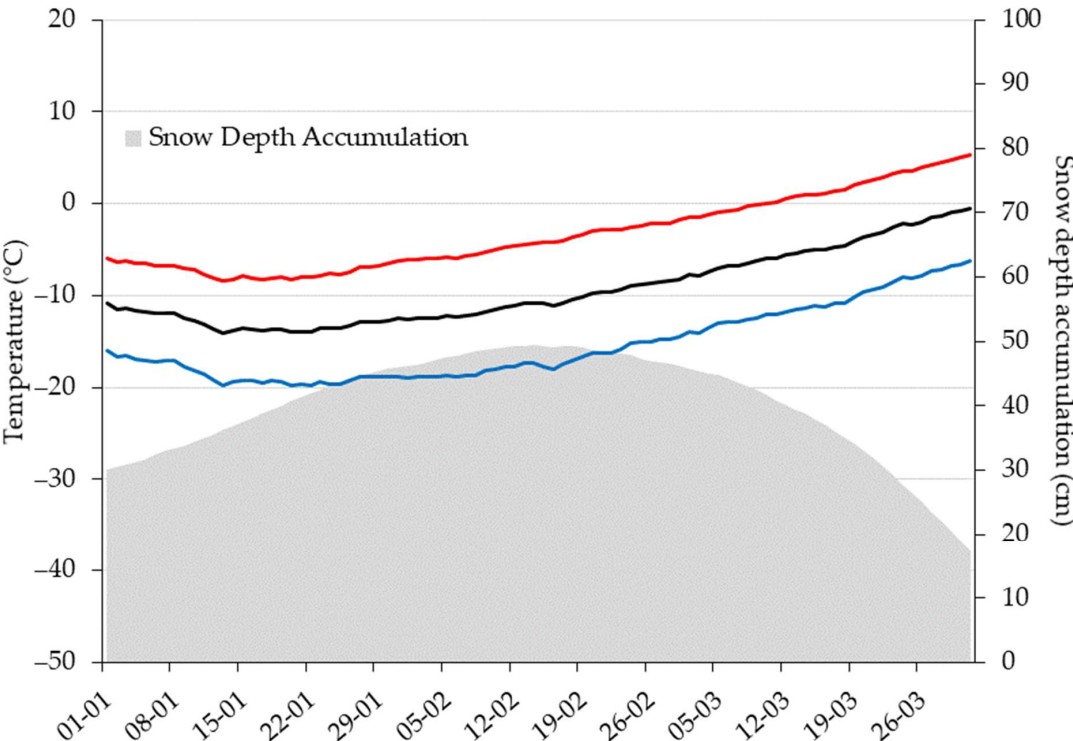

**Figure 2.** Variation of maximum (red), mean (black), and minimum (blue) daily air temperatures (January-March) for climatic normals (1991–2020) at the Grandes-Piles forest nursery (Québec, QC, Canada). The areas in gray indicate snow depth accumulation.

For each sampling period (mid-January and mid-March) and after having thawed the root plugs at 4 °C in the dark, the seedlings were watered by capillarity to standardize the water content of the peat substrate (55% to 60%, *v/v*) and the frost kinetics in the root plugs among seedlings of the same species [36,41]. Then, seedlings of the two forest species were subjected to three treatments at the beginning (mid-January) and end of winter (mid-March):

- Treatment 1 (TR1): Control seedlings were kept in a cold room at 4 °C in the dark (thus simulating light conditions under a relatively deep snow cover).
- Treatment 2 (TR2): Seedlings were placed for 1 day in a greenhouse at 10 °C under winter natural photoperiod conditions.
- Treatment 3 (TR3): Seedlings were placed for 3 days in a greenhouse at 10 °C under winter natural photoperiod conditions.

In eastern Canada (Québec), these warming treatments are considered extreme because the seedlings are without any snow cover, and they are subjected to high day/night temperatures (10 °C) for 1 and 3 days. The succession of these extreme environmental conditions applied at the beginning and the end of winter exceeded the normal climatic conditions (Figure 2) and the projected warming scenario (3 °C to 7 °C) for the province of Québec [43].

For each sampling period (mid-January and mid-March), the seedlings of each species from these three treatments were subjected to different freezing temperatures (control 4 °C, −4 °C, −12° C, and −20 °C) using the same freezing routine described in Section 2.2 and in other publications [36–38]. Although rare and extreme, −20 °C was included as a treatment because this temperature represents the minimum threshold to which seedlings could be subjected in the total absence of snow cover (see climatic normals in Figure 2).

To validate the temperature program for each sampling period and to follow the evolution of the temperatures reached both in the aerial parts and in the root plugs of black spruce (50 cm$^3$) and white spruce (310 cm$^3$), five temperature probes (Model 107B, Campbell Scientific (Edmonton, AB, Canada) Corp., Edmonton, AB, Canada) were used. For each species, 2 probes were inserted in the center of the root plugs. The remaining probe was placed inside the freezer to monitor the air temperature inside the freezer to which seedlings shoots were subjected during the freezing cycle. A datalogger (model CR10X, Campbell Scientific, Canada Corp., Edmonton, AB, Canada) recorded temperatures every three minutes.

For each forest species and each sampling period (mid-January and mid-March), the growth variables and those associated with frost tolerance were evaluated using the same approach described in Section 2.2. The number of plants used, and variables determined were as follows:

- The ratio of dry mass (DM) to fresh mass (FM) of excised terminal shoot tips (4 cm long) was evaluated using 12 seedlings (3 shoot tips per composite sample, 4 composite samples per treatment). This ratio is used to predict frost tolerance in autumn [3,7].
- Electrical conductivity and an index of injury of excised terminal shoot tips of black spruce and white spruce seedlings in response to different artificial freezing temperatures (28 seedlings total; 3 shoot tips per Erlenmeyer flask per temperature per block distributed in four complete random blocks).
- Artificial freezing of whole seedlings for the evaluation of growth variables (dry shoot dry mass, initial roots and new roots, total dry mass), bud burst and recovery (32 seedlings total per treatment per species at the rate of 2 seedlings per temperature per block distributed in four complete random blocks).

### 2.4. Mineral Nutrition

The mineral nutrition of white and black spruce seedlings was assessed in the fall, early, and late winter to determine if the nutritional status of the seedlings varied with the duration of the warm temperatures (1 vs. 3 days). Thus, in addition to the characterization of the mineral nutrition in autumn, we also evaluated the mineral nutrition of seedlings

that underwent the same treatments (TR1, TR2, and TR3) at the beginning and end of winter. The mineral nutrient concentrations and contents (N, P, K, Ca, and Mg) of the shoots, roots, and whole seedlings of black and white spruce for each treatment were determined using 4 composite samples (10 seedlings per composite sample per block). The mineral content was calculated for each nutrient (concentration multiplied by dry mass) [44]. Mineral nutrient analyses of seedlings were determined at the organic and inorganic chemistry laboratory of the Québec Forest Research Branch using the methods described in our previous studies [28–30].

### 2.5. Statistical Analyses

An analysis of variance was performed using mixed linear models with the MIXED procedure of SAS/STAT version 14.1 (SAS Institute Inc. Cary, NC, USA, 2015). In all models, the number of degrees of freedom of the denominator for fixed effects tests was calculated using the Kenward-Roger method.

During the various artificial freezing treatments, the seedlings were kept inside the freezer, and they were taken out for one hour after the target temperature was reached. This implied that no randomization was performed when applying the different freezing treatments to assess the different cold tolerance variables. Thus, the statistical model used was a strip-plot, also known as a split-block [45,46]. The model's random part was simplified as described in Bernier-Cardou and Bigras [45] at the 30% threshold, and the significance thresholds for the fixed effects were set at 5%.

To assess the effects of forest species and different artificial freezing temperatures (control 4 °C, −5 °C, −10 °C, −15 °C and −20 °C) on relative electrical conductivity, index of injury, and different growth variables at the end of the fall, forest species, and freezing temperatures, and their interactions were considered fixed effects, while the block and the interaction between the block and factors were considered as random effects.

For frost tolerance in response to different periods of warm weather at the beginning and end of winter, sampling period, treatments at the beginning and end of winter, freezing temperatures (control 4 °C, −4 °C, −12° C and −20 °C), and their interaction were considered as fixed effects, while the block and the interaction between the block and the factors were considered as random effects.

A model with a fixed effects sampling period, treatments at the beginning and end of winter, and their interaction and a random block in the sampling period effect was used to analyze the ratio of dry mass to fresh mass.

To assess the effects of the sampling period and treatments at the beginning and end of winter on nutrient concentrations and the contents of different parts of spruces before freezing, a means model was used to account for the control in autumn measures and analyze the two-way structures with missing treatment combinations. A factor combining the sampling period and treatments was created, and tests and comparisons were made using the contrasts.

For each of the analyses performed, the assumption of normality and homogeneity of variances were graphically checked. To account for the heterogeneity of the variances, the residual variance was weighted according to the variance observed for each forest species for the analysis at the end of the fall. For the other analysis, the models were adjusted to the dataset for the different forest species, since our objective was not to compare the state of the hardening of the seedlings between the two species.

Subsequently, multiple comparisons of the means were performed by forest species when a fixed effect was significant. Multiple comparison thresholds are adjusted with a simulation method available in SAS (SAS Institute Inc. Cary, NC, USA, 2015) to detect significant differences between means, as described by Westfall et al. [47].

## 3. Results

*3.1. Assessment of Frost Tolerance of One-Year-Old Black Spruce (1 + 0) and Two-Year-Old White Spruce (2 + 0) Seedlings at the End of the Fall*

At the end of the fall (November 30), no significant difference was observed between the mean relative electrical conductivities and the index of injury of the shoots of the two forest species) in response to the different freezing temperatures tested (Table 2). Thus, for example, the average values of the relative electrical conductivities of black spruce (1 + 0) at temperatures of 4 °C and −20 °C were 3.40 ± 0.20% and 3.58 ± 0.20%, respectively. In the case of white spruce, the averages for these two temperatures (4 °C and −20 °C) were the same (Table 2).

**Table 2.** Comparison of the adjusted means (±standard error) of the relative electrical conductivity (RC) and the index of injury (It) of white spruce (WS) and black spruce (BS) seedlings in response to different freezing temperatures at the end of autumn (30 November).

|  |  | **−20 °C** | **−15 °C** | **−10 °C** | **−5 °C** | **4 °C** |
|---|---|---|---|---|---|---|
| RC (%) | WS (2 + 0) | 2.48 ± 0.20 a [1] | 2.56 ± 0.20 a | 2.42 ± 0.20 a | 2.75 ± 0.20 a | 2.48 ± 0.20 a |
|  | BS (1 + 0) | 3.58 ± 0.20 a | 3.77 ± 0.20 a | 4.09 ± 0.20 a | 3.19 ± 0.20 a | 3.40 ± 0.20 a |
| It | WS (2 + 0) | 0.00 ± 0.12 a | 0.08 ± 0.12 a | −0.06 ± 0.12 a | 0.27 ± 0.12 a |  |
|  | BS (1 + 0) | 0.18 ± 0.35 a | 0.37 ± 0.35 a | 0.71 ± 0.35 a | −0.23 ± 0.35 a |  |

[1] Horizontally, the adjusted means (±standard error) followed by similar letters did not show significant differences at the threshold $\alpha$ = 5%, according to a simulation method available in SAS as described by Westfall et al. [47].

The index of injury (It) calculated for each of the forest species in response to the different freezing temperatures tested was always less than one and did not exceed the maximum value of 0.71 for black spruce and 0.27 for white spruce. This indicates that the seedlings of the two species tolerate the different freezing temperatures tested (T1 = −5 °C, T2 = −10 °C, T3 = −15 °C and T4 = −20 °C) at the end of the fall.

*3.2. Initiation of New Roots and Bud Break in the Case of Whole-Seedling Freeze Treatments at the End of the Fall*

After undergoing the various artificial freezing treatments and after potting the seedlings, followed by a period of 21 days of growth in a greenhouse under optimal environmental conditions, the buds of the main stems and lateral branches of all the spruce seedlings of black spruce (1 + 0) and white spruce (2 + 0) showed bud break (Figures 3 and 4). All the buds and needles showed no sign of browning or mortality, whatever the frost temperature (−5 °C, −10 °C, −15 °C, and −20 °C) (Figures 3 and 4).

For all freezing temperatures tested, all observed seedlings of the two tree species initiated new white roots (Figures 3 and 4, Table 3). However, compared with the control temperature of 4 °C, the frost temperature of −20 °C significantly reduced the new root dry mass in white spruce and black spruce by 44% and 58%, respectively (Table 3). At the end of the bioassay test period (21 days), there was no significant difference in the dry mass of new white roots between the other frost temperatures tested (−5 °C, −10 °C, and −15 °C) and the control temperature (4 °C) (Table 3). With the exception of total and new root dry masses in black spruce, all other growth variables were not significantly affected by the artificial freezing temperatures tested (Table 3).

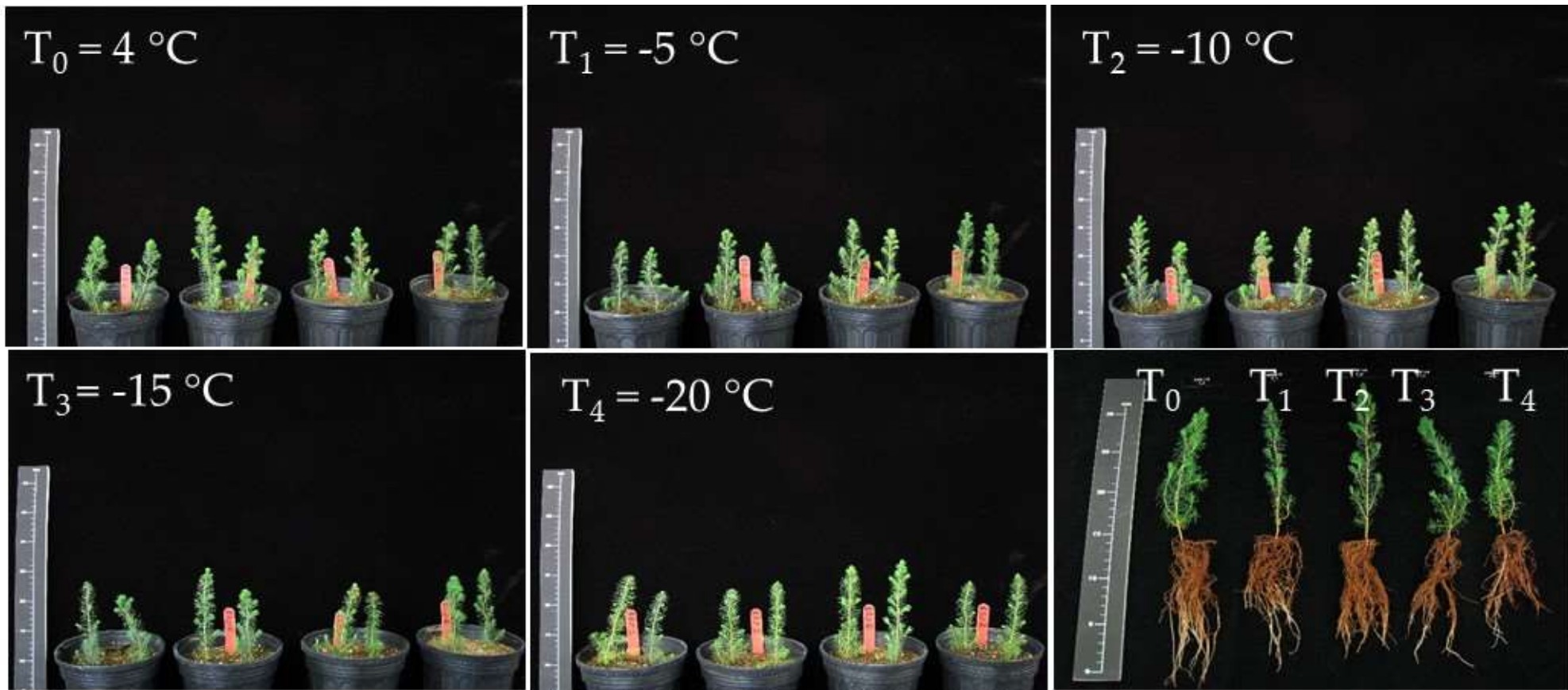

**Figure 3.** Bud burst of all buds of main stems and lateral branches of all black spruce (1 + 0) seedlings in response to different temperatures ($T_0$, $T_1$, $T_2$, $T_3$, and $T_4$) and after 21 days of growth during the recovery period under optimal greenhouse growing conditions. Example of new root initiation in response to the same temperatures and optimal growing conditions.

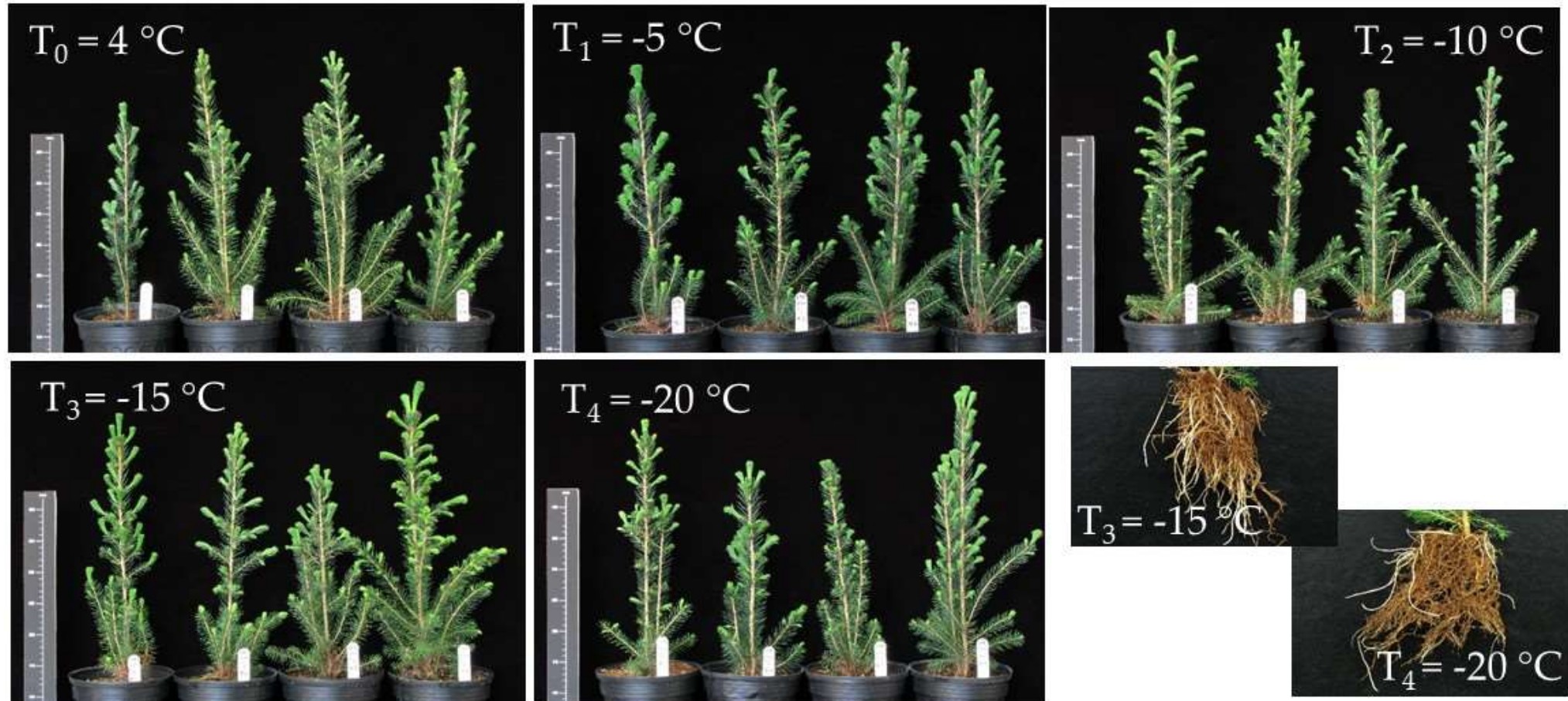

**Figure 4.** Bud burst of all buds of main stems and lateral branches of all white spruce (2 + 0) seedlings in response to different temperatures ($T_0$, $T_1$, $T_2$, $T_3$, and $T_4$) and after 21 days of growth during the recovery period under optimal greenhouse growing conditions. Example of new root initiation in response to the two freezing temperatures ($T_3$ and $T_4$) and optimal growing conditions.

**Table 3.** Comparison of the adjusted means (± standard error) of different growth variables of white spruce (2 + 0) and black spruce (1 + 0) seedlings at the end of the fall in response to different freezing temperatures and after 21 days of growth during the recovery period under optimal greenhouse growing conditions.

| | Black Spruce (1 + 0) | | | | | White Spruce (2 + 0) | | | | |
|---|---|---|---|---|---|---|---|---|---|---|
| | $T_0 = 4\,°C$ | $T_1 = -5\,°C$ | $T_2 = -10\,°C$ | $T_3 = -15\,°C$ | $T_4 = -20\,°C$ | $T_0 = 4\,°C$ | $T_1 = -5\,°C$ | $T_2 = -10\,°C$ | $T_3 = -15\,°C$ | $T_4 = -20\,°C$ |
| Dry mass of new white roots (mg) | 37.5 ± 5.1 a [1] | 38.3 ± 5.1 a | 27.3 ± 5.1 ab | 18.4 ± 5.1 ab | 15.9 ± 5.1 b | 62.4 ± 15.14 a | 85.8 ± 15.14 a | 79.6 ± 15.14 a | 85.1 ± 15.14 a | 35.1 ± 15.14 b |
| Initial root dry mass (mg) | 251.0 ± 18.3 a | 216.3 ± 18.3 a | 236.8 ± 18.3 a | 196.9 ± 18.3 a | 186.8 ± 18.3 a | 1989.5 ± 248.1 a | 2330.0 ± 248.1 a | 1958.3 ± 248.1 a | 2596.4 ± 248.1 a | 1987.9 ± 248.1 a |
| Total root dry mass (mg) | 288.5 ± 22.8 a | 254.5 ± 22.8 ab | 264.0 ± 22.8 ab | 215.3 ± 22.8 ab | 202.6 ± 22.8 b | 2051.9 ± 257.8 a | 2415.8 ± 257.8 a | 2037.9 ± 257.8 a | 2681.5 ± 257.8 a | 2023.0 ± 257.8 a |
| Shoot dry mass (g) | 0.67 ± 0.05 a | 0.60 ± 0.05 a | 0.62 ± 0.05 a | 0.53 ± 0.05 a | 0.58 ± 0.05 a | 10.16 ± 0.98 a | 11.57 ± 0.98 a | 9.61 ± 0.98 a | 11.95 ± 0.98 a | 10.16 ± 0.98 a |
| Total dry mass (g) | 0.96 ± 0.06 a | 0.86 ± 0.06 a | 0.88 ± 0.06 a | 0.74 ± 0.06 a | 0.79 ± 0.06 a | 12.22 ± 1.19 a | 13.98 ± 1.19 a | 11.65 ± 1.19 a | 14.63 ± 1.19 a | 12.18 ± 1.19 a |

[1] Horizontally, the adjusted means (±standard error) for each species followed by similar letters did not show significant differences at the threshold $\alpha = 5\%$, according to a simulation method available in SAS, as described by Westfall et al. [47].

### 3.3. Frost Tolerance of Black Spruce (1 + 0) and White Spruce (2 + 0) Seedlings in Response to Different Periods of Warm Weather at the Beginning and End of Winter

3.3.1. Determination of Shoot Cold Tolerance of Black Spruce and White Spruce Using Electrolyte Conductivity Measurements in Response to Different Periods of Warm Weather at the Beginning and End of Winter

The analysis of variance showed that there was a significant effect from the sampling period on the relative electrical conductivity of the shoots of white spruce ($p = 0.0427$) and a significant interaction between the sampling period and the treatment ($p = 0.0409$) (Table 4). We also observed a significant effect of date and treatment for the injury index in black spruce ($p = 0.0391$).

**Table 4.** Observed probabilities (Pr > F) and degrees of freedom of the fixed effects associated with the analysis of variance of the relative electrical conductivity (RC) and index of injury (It) of shoots of two-year-old white spruce (WS, 2 + 0) and one-year-old black spruce (BS, 1 + 0).

| | | Relative Electrical Conductivity (%) | | | | | Index of Injury | | | |
| | | WS 2 + 0 | | BS 1 + 0 | | | WS 2 + 0 | | BS 1 + 0 | |
| Source of Variation | dln [1] | dld | Pr > F | dld | Pr > F | dln | dld | Pr > F | dld | Pr > F |
|---|---|---|---|---|---|---|---|---|---|---|
| Period | 1 | 6 | 0.0427 | 6 | 0.0778 | 1 | 18 | 0.9492 | 18 | 0.0391 |
| Treatment | 2 | 12 | 0.1866 | 12 | 0.0139 | 2 | 18 | 0.0557 | 18 | 0.0391 |
| Period × Treatment | 2 | 12 | 0.5206 | 12 | 0.0409 | 2 | 18 | 0.2919 | 18 | 0.7038 |
| Temperature | 3 | 54 | 0.1098 | 54 | 0.4875 | 2 | 36 | 0.2587 | 36 | 0.6295 |
| Period × Temperature | 3 | 54 | 0.5516 | 54 | 0.1045 | 2 | 36 | 0.3411 | 36 | 0.6515 |
| Treatment × Temperature | 6 | 54 | 0.2293 | 54 | 0.0636 | 4 | 36 | 0.9011 | 36 | 0.3974 |
| Period × Treatment × Temperature | 6 | 54 | 0.5235 | 54 | 0.3738 | 4 | 36 | 0.6630 | 36 | 0.2057 |

[1] dln: degrees of freedom of numerator; dld: degrees of freedom of denominator.

The relative electrical conductivity of the shoot of white spruce in mid-March was significantly higher than in mid-January (Table 5). On the other hand, in the case of black spruce, the relative electrical conductivity of the shoots of the control seedlings maintained at 4 °C in the dark was significantly lower than the treatment of exposure for 3 days at 10 °C for the mid-March sampling period (Table 5). The injury index of the shoots of black spruce in mid-March was significantly higher than that of January (Table 5).

**Table 5.** Comparison [1] of adjusted means of the relative electrical conductivity (RC) and Index of Injury (It) of shoots of two-year-old white spruce and one-year-old black spruce seedlings between sampling date at the beginning (mid-January) and end of winter (mid-March) and three treatments (Control at 4 °C in dark conditions, 1 day at 10 °C and 3 days at 10 °C under the natural photoperiod of the season).

| RC (%) | WS 2 + 0 | Mid-January | | | Mid-March | | | | | | | |
|---|---|---|---|---|---|---|---|---|---|---|---|---|
| | | 2.94 | ±0.15 | b [2] | 3.49 | ±0.15 | a | | | | | |
| | BS 1 + 0 | | Mid-January | | | | | | Mid-March | | | |
| | | One day | | 3 days | | Control | | One day | | 3 days | | Control |
| | | 3.24 ±0.23 ab | | 3.22 ±0.23 ab | | 3.13 ±0.23 b | | 3.77 ±0.23 ab | | 4.25 ±0.23 a | | 3.38 ±0.23 b |
| It | BS 1 + 0 | One day | | 3 days | | Control | | | | | | |
| | | 0.30 ±0.18 a | | 0.34 ±0.18 a | | −0.28 ±0.18 a | | | | | | |
| | | Mid-January | | Mid-March | | | | | | | | |
| | | −0.11 ±0.14 b | | 0.35 ±0.14 a | | | | | | | | |

[1] Comparisons are made only when the probabilities are significant. For more details, see Table 4. [2] Horizontally, the adjusted means (±standard error) followed by distinct letters show significant differences at the threshold $\alpha = 5\%$, according to a simulation method available in SAS as described by Westfall et al. [47].

### 3.3.2. Determination of Whole-Seedling Freezing Temperature Effects on the Growth of New Roots, Shoot, and Bud Breaks in Response to Different Periods of Warm Weather at the Beginning and End of Winter

The monitoring of the different temperatures inside the freezer and in the root plugs of the white and black spruce seedlings is shown in Figure 5. Shoots of the two species were gradually exposed to the different target freezing temperatures ($-4$ °C, $-12$ °C, and $-20$ °C). In the case of root plugs, the cooling kinetics evolved differently depending on the volume of the root plug (50 cm$^3$ or 310 cm$^3$) (Figure 5) despite that the substrate water contents were high (55 to 60%, *v/v*) and similar between the two containers (50 cm$^3$ or 310 cm$^3$) before placing the seedlings in the freezer. The temperature of $-20$ °C in the root plugs of white spruce (2 + 0) was reached almost at the same time as in the root plugs of black spruce (50 cm$^3$), but the transfer kinetics of frost in the two types of root plugs were different (Figure 5).

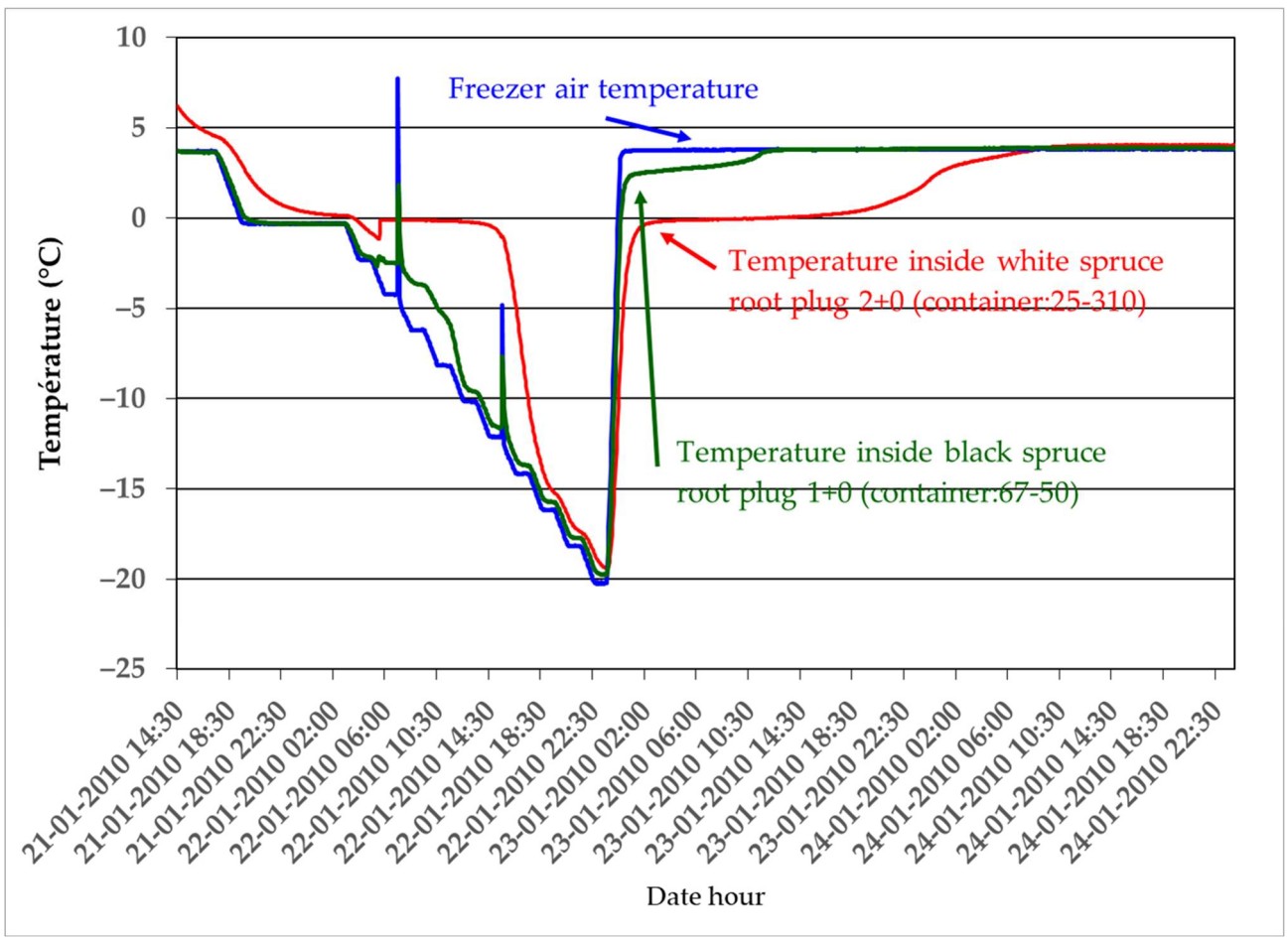

**Figure 5.** Temperatures registered in the air around the shoots and in the root plugs of black spruce (1 + 0) and white spruce (2 + 0) during artificial frost treatments. The seedlings were removed from the freezer once the target temperature around shoots ($-4$ °C, $-12$ °C, and $-20$ °C) had been reached and maintained for one hour. The peaks in blue indicate when the freezer door was opened to remove the seedlings.

The sampling period had a significant effect on the ratio of dry mass to fresh mass for the white spruce (2 + 0) ($p = 0.0270$) and black spruce (1 + 0) ($p < 0.0001$). On the other hand, the effects of the treatment in relation to the exposure of the seedlings (4 °C in darkness, 1 day, and 3 days exposed to 10 °C) and the interaction sampling period × treatment was not significant ($p > 0.15$). Thus, the ratios determined in mid-March, regardless of the forest

species (white spruce: 43.71 ± 0.18; black spruce: 36.67 ± 0.16), were significantly lower than those determined in mid-January (white spruce: 44.33 ± 0.18; black spruce: 38.18 ± 0.16).

After being exposed to the different freezing temperatures and after 21 days under optimal growth conditions in the greenhouse, the analysis of variance of the growth variables showed significant effects of the sampling period × temperature interaction for shoot dry mass, dry mass of new roots for white spruce, total root dry mass for black spruce, and total dry mass of seedling, as well as simple significant effects of temperature for the dry mass of new roots for black spruce and total root dry mass for white spruce and sampling period for the dry mass of new roots for black spruce (Table 6). In the case of white spruce (2 + 0), −20 °C in mid-March significantly reduced the growth of new roots compared to the other freezing temperatures (Figure 6). The comparisons of the means for the other growth variables after the period of recovery under optimal growth greenhouse conditions following the different freezing temperatures are indicated in Table S1.

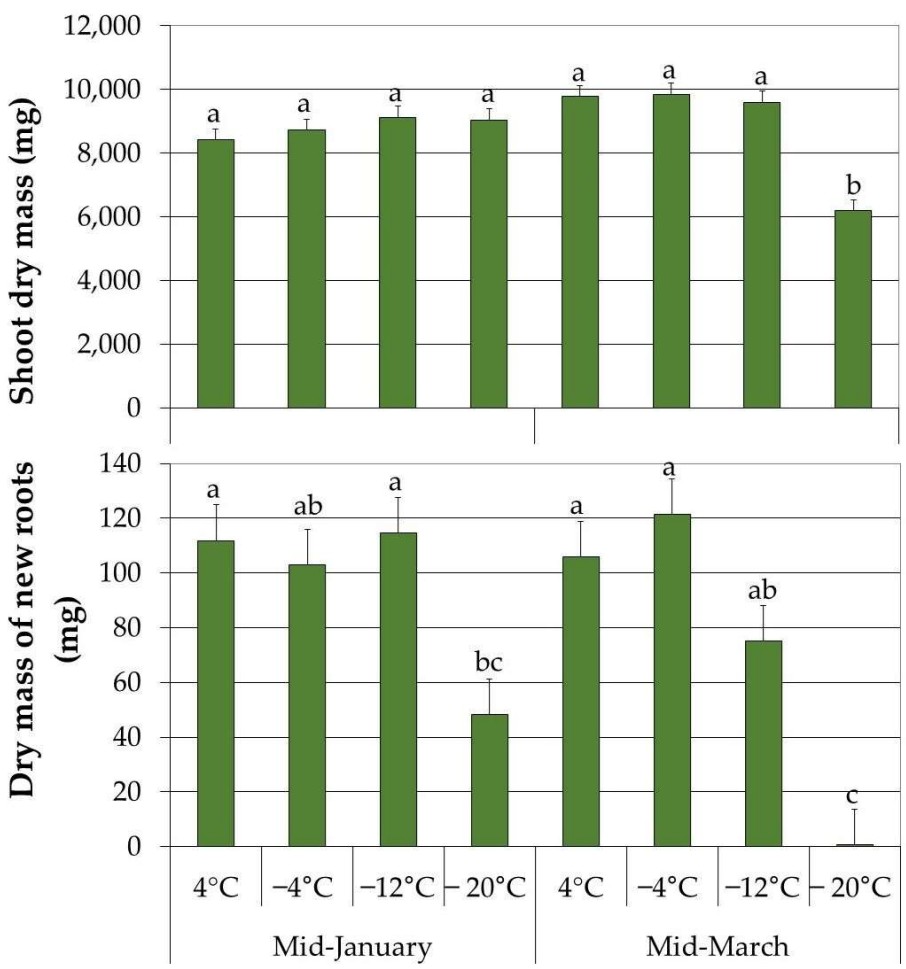

**Figure 6.** Comparisons of adjusted means (±standard error) of dry masses of shoot and new roots of white spruce (2 + 0) in response to whole-seedling freezing temperatures at the beginning (mid-January) and end of winter (Mid-March). The distinct letters show significant differences at the threshold α = 5%, according to a simulation method available in SAS as described by Westfall et al. [47].

**Table 6.** Observed probabilities (Pr > F) and degrees of freedom of the fixed effects associated with the analysis of variance of different growth variables of two-year-old white spruce (WS, 2 + 0) and one-year-old black spruce (BS, 1 + 0) seedlings at the beginning (mid-January) and end of winter (mid-March).

| Source of Variation | dln [1] | Shoot Dry Mass (mg) | | | | Dry Mass of New Roots (mg) | | | | Total Root Dry Mass (mg) | | | | Total Dry Mass of Seedling (mg) | | | |
|---|---|---|---|---|---|---|---|---|---|---|---|---|---|---|---|---|---|
| | | WS 2 + 0 | | BS 1 + 0 | | WS 2 + 0 | | BS 1 + 0 | | WS 2 + 0 | | BS 1 + 0 | | WS 2 + 0 | | BS 1 + 0 | |
| | | dld | Pr > F | dld | Pr > F | dld | Pr > F | dld | Pr > F | dld | Pr > F | dld | Pr > F | dld | Pr > F | dld | Pr > F |
| Period | 1 | 18 | 0.9184 | 21.4 | 0.4062 | 72 | 0.0401 | 24 | 0.0002 | 18 | 0.2280 | 6 | 0.0762 | 18 | 0.7931 | 20.1 | 0.8152 |
| Treatment | 2 | 18 | 0.8173 | 14.3 | 0.1117 | 72 | 0.5013 | 48 | 0.4314 | 18 | 0.4471 | 66 | 0.1280 | 18 | 0.6680 | 13.6 | 0.0993 |
| Period × Treatment | 2 | 18 | 0.6220 | 14.3 | 0.2499 | 72 | 0.9538 | 48 | 0.4529 | 18 | 0.2910 | 66 | 0.0604 | 18 | 0.8633 | 13.6 | 0.1743 |
| Temperature | 3 | 54 | <0.0001 | 19.2 | 0.0004 | 72 | <0.0001 | 24 | 0.0002 | 54 | <0.0001 | 66 | 0.0005 | 54 | <0.0001 | 18.6 | 0.0005 |
| Period × Temperature | 3 | 54 | <0.0001 | 19.2 | 0.0012 | 72 | 0.0429 | 24 | 0.1977 | 54 | 0.0666 | 66 | 0.0006 | 54 | <0.0001 | 18.6 | 0.0012 |
| Treatment × Temperature | 6 | 54 | 0.2458 | 36.9 | 0.5266 | 72 | 0.3370 | 48 | 0.1401 | 54 | 0.5086 | 66 | 0.8650 | 54 | 0.3127 | 38.2 | 0.6768 |
| Period × Treatment × Temperature | 6 | 54 | 0.5690 | 36.9 | 0.7818 | 72 | 0.4332 | 48 | 0.8765 | 54 | 0.5230 | 66 | 0.9975 | 54 | 0.6289 | 38.2 | 0.9153 |

[1] dln: degrees of freedom of numerator; dld: degrees of freedom of denominator.

At the beginning of winter, all the buds of the main stems and branches of the white and black spruce seedlings of these two treatments (1 day at 10 °C and 3 days at 10 °C in the natural photoperiod of the season) showed all the normal bud break similar to the control seedlings (Figure 7). All seedlings also showed normal growth (Figure 7). On the other hand, seedlings sampled in late winter (mid-March) showed that those subjected to −20 °C did not bud break compared with the seedlings of the other warming treatments subjected to other temperatures (4 °C, −4 °C and −12 °C) (Figure 8). A few needles and buds of some of the seedlings subjected to −20 °C died. Thus, the dry shoot masses of seedlings subjected to −20 °C in mid-March were significantly lower than the other combinations of freezing temperatures and the sampling period (Figures 6 and 8).

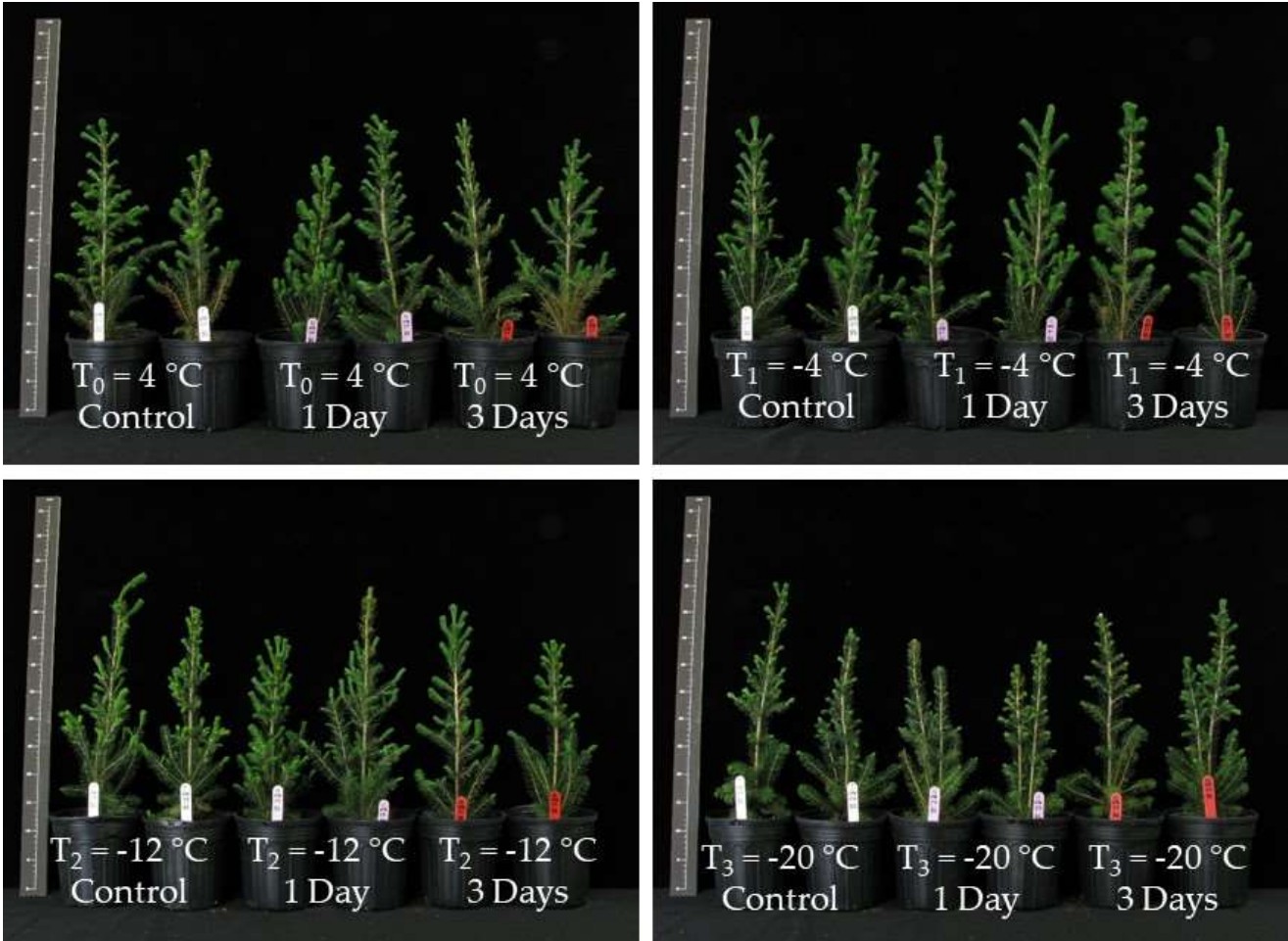

**Figure 7.** Normal bud break of the apical buds and lateral branches of white spruce seedlings (2 + 0) in response to their exposure to the three treatments (Control at 4 °C in the dark, 1 day at 10 °C and 3 days at 10 °C under the natural photoperiod of the season) and followed by artificial freezing temperatures (−4 °C, −12 °C and −20 °C) at the beginning of winter (mid-January).

### 3.3.3. Mineral Nutrition in Autumn, Winter and towards the End of Winter in Response to Warming Treatments

In the case of the shoots of black spruce seedlings (1 + 0), the warming treatments had a significant effect on the concentration of nitrogen ($p = 0.0462$), potassium ($p = 0.0156$), calcium ($p = 0.0249$) and magnesium ($p = 0.0196$) but no effect on phosphorus concentration ($p = 0.4650$; Table S2). However, the same treatments had no significant effect on the concentration of all the mineral elements in shoots of white spruce seedlings (2 + 0, $p > 0.08$; Table S2). Probabilities

of significance for root and whole seedling mineral nutrient concentrations and contents are shown in Table S2.

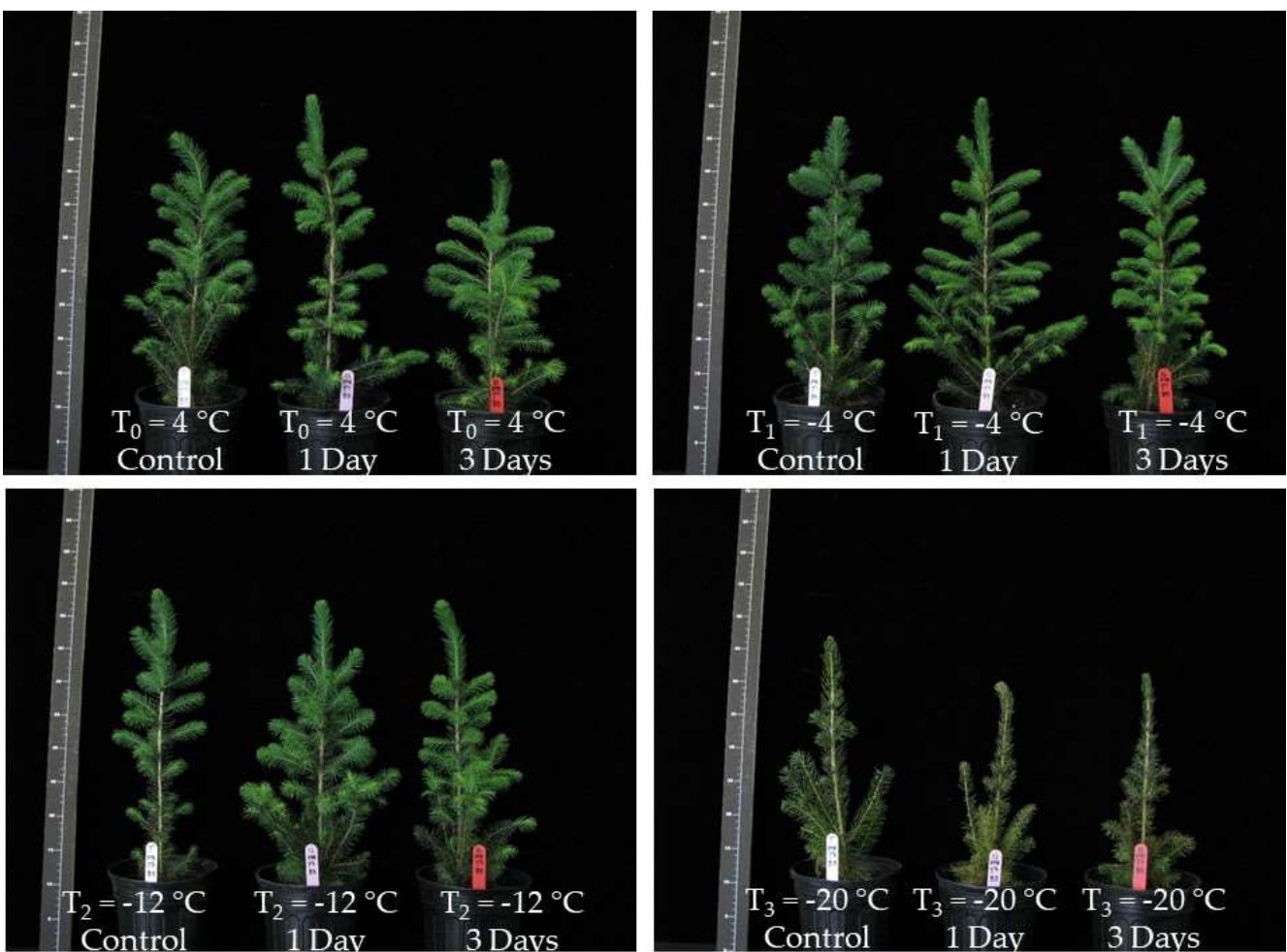

**Figure 8.** Normal bud break of the apical buds and lateral branches of white spruce seedlings (2 + 0) at the end of winter (mid-March) in response to their exposure to the three warming treatments (control at 4 °C in the dark conditions, 1 day at 10 °C, and 3 days at 10 °C under the natural photoperiod of the season) and followed by artificial freezing temperatures ($-4$ °C and $-12$ °C). Conversely, seedlings subjected to the $-20$ °C freezing temperature and the same warming treatments showed no bud burst.

Additionally, there was no significant difference in shoot nitrogen concentration of the black spruce and white spruce seedlings among the treatments (initial shoot nitrogen concentration in autumn, TR1, TR2 and TR3; Table S3), regardless of the sampling period. Thus, for example, in the case of black spruce, the average shoot nitrogen concentration for the three treatments at the beginning and at the end of winter varied between $1.76 \pm 0.05\%$ and $1.96 \pm 0.05\%$. In the case of white spruce, it varied between $1.93 \pm 0.07\%$ and $2.06 \pm 0.07\%$ (Table S3). The absence of significant difference was also observed in the nitrogen content of the whole seedlings and the roots (Table S3). Thus, warming treatments at the beginning and the end of winter did not induce significant changes in mineral nitrogen nutrition (Table S3).

## 4. Discussion

Our results revealed that the acquisition of frost tolerance in the fall (Table 2, Figures 3 and 4) combined with the presence of a layer of snow in the forest nursery enabled the seedlings to withstand climatic extremes characterized by mild spells followed

by freezing temperatures in winter (Figure 7). On the other hand, the decrease in snow cover depth at the end of winter (Figure 1b) and the exposure of seedlings to periods of realistic mild spells, as predicted by climate change scenarios in eastern Canada [42], followed by freezing temperatures induced the loss of cold hardiness at −20 °C, but not at −4 °C and −12 °C (Figure 8).

The winter survival capacity of seedlings of boreal forest species produced in forest nurseries depends not only on their initial level of cold tolerance at the end of autumn, which is influenced by cultural practices [2,7,8], but also on their ability to maintain a level of cold hardiness in response to mild spells in winter followed by freezing temperatures. To ensure that the seedlings of the two forest species (black spruce 1 + 0 and white spruce 2 + 0) are well hardened at the end of autumn, we measured the relative electrical conductivity and index of injury using 4 cm long apical shoot tips and found no significant difference in all freezing temperatures tested (Table 2). This was also verified using whole seedlings, which were subjected to freezing temperatures followed by a recovery period under optimal environmental greenhouse conditions for growth. All buds of the seedlings of the two forest species showed bud break, good growth of their shoots, and no apparent freezing damage (Figures 3 and 4). These results and observations are similar to those obtained on black and white spruce in autumn in eastern Canada [3,4,36,48].

However, in late autumn, the roots of both forest species survived and showed less growth at −20 °C compared with other freezing temperatures recorded in root plugs. At this time of year (November 30), the −20 °C temperature observed in root plugs was unlikely in eastern Canada [3]. On the other hand, shoots of all white and black spruce seedlings that were subjected to −20 °C were not affected and showed normal budburst and growth (Figures 3 and 4). This clearly indicates that the kinetics of hardening and the tolerance to freezing temperatures of the shoots and roots are not synchronized with a delay in root hardening, as observed in other studies [2,7,8]. This delay in the hardening of the roots of the two tree species in the forest nursery is largely due to the active growth of the root system in late autumn and to the non-freezing temperatures of the root plugs, which are favorable to their growth [7,8]. In fact, after the growth of the shoots has stopped and buds have formed, the dry root mass of white and black spruces doubles in autumn under forest nursery conditions [41,49]. The monitoring of temperatures during a period in autumn (1 September–10 November) in six forest nurseries in Québec showed that the minimum temperatures in the root plugs of white spruce seedlings (1 + 0) rarely dropped below −6 °C, whereas those of the air could reach −12 °C [3].

The improvement in hardening and frost tolerance of black and white spruce seedlings in fall in eastern Canada under forest nursery conditions is the result of the success of different cultural practices, in particular, the control of irrigation [41,49] and fertilization [50] according to the plant growth stages, the accumulation of hardening degree days [48], the imposition of water stress [7,51,52], short-day treatments [2,8,28] to induce bud formation and epicuticular wax formation, and the increase of the ratio of dry mass to fresh mass (DM/FM) of the apical shoots greater than 30% [7,48,53,54]. Our results showed that these ratios reached 43.04% and 46.63% for black spruce and white spruce, respectively (Table 1). By using whole seedlings to assess frost tolerance, it was shown that DM/FM values similar to those obtained in our study indicate that the shoots can tolerate −20 °C without any apparent damage [3,48]. Despite the lack of consensus regarding the effects of high foliar concentrations of mineral nutrients on the cold tolerance of seedlings [7], foliar nitrogen concentrations in the two tree species were optimal in autumn (1.99 to 2.06%, Table S3) and in winter (1.76 to 2.06%, Table S3) and they did not negatively affect the cold tolerance of shoots even for the temperature of −20 °C. These results corroborate those of our previous studies, which showed no negative effect on the optimal foliar nitrogen concentrations (1.63% to 2.41%) of white spruce seedlings (1 + 0) on their frost tolerance in autumn under six forest nursery conditions [3].

Our results showed that winter warming treatments applied in mid-January (1 day at 10 °C or 3 days at 10 °C) did not reduce the cold hardiness of black and white spruce

seedlings and these responses are different from those reported in [19] for the same species. These different results could be due to four main factors, including the use of different seed genetic sources [55], cultural practices for growing seedlings [8], their preconditioning to frost [8], and the duration and temperature of warming [56]. Man et al. [19] applied long durations of warming (5, 10, and 15 days) and a high temperature (16 °C), which are far from climatic normals at our forest nursery (Figure 2). Moreover, the warming scenario applied in terms of duration and intensity by Man et al. [19] suggests that seedlings become more susceptible to winter frost. This warming scenario remains improbable in forest nurseries in Québec based on the warming scenario (3 °C to 7 °C) projected by Ouranos for the province of Québec [43]. However, at the end of winter (mid-March), the control seedlings and those of warming treatments subjected to −20 °C did not break bud and did not develop new roots compared with the seedlings of the same treatments subjected to other temperatures (4 °C, −4 °C and −12 °C) (Figure 8). Thereby, the hypothesis that black and white spruce seedlings produced in forest nurseries in eastern Canada rapidly lose their cold hardiness in response to mild winter spells is partially true, especially towards the end of the winter (mid-March). This also clearly indicates that the cold hardiness to −20 °C of white and black spruce control seedlings in mid-March was lost and did not remain constant over the winter (based on mid-January measurements), which corroborates the results from other studies [57,58]. For instance, Tumanov and Krasavtsev [58] showed that under controlled conditions, the maximum level of cold hardiness of northern woody plants could be lost in a few hours when the air temperature increases above 0 °C. The rapid loss of frost tolerance observed in both species at the end of winter when the temperature was −20 °C is probably due to the increase in air temperature (>5 °C) during this exceptional year (Figure 1b), as reported in other studies [59,60]. However, our results are not in agreement with previous studies [61] suggesting that cold hardiness cannot be rapidly lost in late winter to early spring. On the other hand, the survival of the other seedlings of the same treatments, which were subjected to other freezing temperatures (−4 °C and −12 °C), were not negatively affected by these frost levels and the seedlings showed normal growth (bud break, etc.) like the control seedlings (Figure 8). The analysis of climate normals at the Grandes-Piles forest nursery showed that the minimum temperature of −20 °C was rare during the month of March (Figure 2), but for certain exceptional years, the frequency is high (Figure 1a) or rare (Figure 1b). These results suggest that the combination of the early and significant decrease in the depth of snow protecting the seedlings and the periods of warm spells followed by severe freezing temperatures (< −20 °C) at the end of winter or at the beginning of spring during exceptional climatic years will contribute to an increased loss of seedlings in northern forest nurseries.

Significant containerized seedlings losses caused by root frost are very common in forest nurseries in Nordic countries that overwinter outdoors [62]. The absence of initiation of new roots and bud bursts towards the end of winter (mid-March) in the white and black spruce seedlings of the three treatments (Figures 6 and 8) subjected to a temperature of −20 °C is due to root-freezing damages as observed in other studies for the same species [62,63]. Roots damaged by frost cause a significant reduction in the absorption of water and mineral elements and net photosynthesis, as well as survival and growth in reforestation sites [62,63]. This reduction in photosynthesis negatively affects the growth of new roots because the latter is intimately linked to the products of current photosynthates from the shoot in conifer seedlings [64–66]. Carles et al. [66] showed that the growth of new roots during a 21-day period under optimal greenhouse environmental conditions following freezing treatments was negatively related to the proportion of damaged needles and positively linked to the photosynthetic capacity of living needles of white spruce seedlings.

Although several techniques have been developed to determine root freezing damage [62], symptoms of root frost damage are not easily detectable by nursery managers on shoots before bud break. Rapid detection of root frost is of great interest to forest nursery managers to avoid reforestation of damaged seedlings whose survival is compromised.

Conversely, the initiation and growth of new roots, bud breaks, and elongation of main stems and branches are good indicators that integrate both the survival and physiological functioning of the seedling following warming and freezing temperature treatments applied in late fall, early, and late winter (Figures 3, 4 and 6–8). In addition, the absence of visible damage on the shoots in response to warming treatments and freezing temperatures clearly shows the frost tolerance of white and black spruce seedlings. This indicates that the physiological processes (gaz exchange parameters, water and mineral absorption, etc.) of these two spruce species quickly respond during the recovery period following freezing temperatures. Merry et al. [67] showed that the photosynthetic function of white spruce could recover up to three times more rapidly than that of Eastern white pine in response to freezing temperatures during winter.

To avoid any risk associated with climatic extremes in winter, several Nordic nurseries place seedlings in frozen storage during winter (−2 °C to −5 °C) [2,8,68,69]. On the other hand, to reduce the loss of seedlings caused by the different types of frost (early frost, late frost, root frost, winter desiccation, etc.) when seedlings overwinter outside, the nursery managers do not have the choice to use different cultural practices and protection strategies to improve cold hardiness of seedlings. Thus, acquiring and improving the frost tolerance of seedlings in autumn is an essential prerequisite for survival during winter. This improvement in frost tolerance in autumn can be obtained by resorting to a succession of different cultural practices, such as lowering the water content of the substrate from 15% to 20% (*v/v*) to induce water stress [6,40,48,50,51], reducing the nitrogen fertility of the growing media (e.g., black spruce 1 + 0:25 ppm; white spruce 2 + 0:50 ppm) [49], and applying a short-day treatment [6,7,27]. Before the arrival of the first frost temperatures, the water content of the substrate must be increased (50 to 60%, *v/v*) because moist substrates release heat compared with dry substrates, which offers better protection to the roots against frost. Another cultural practice of overwintering seedlings (1 + 0) is used by directly laying protective covers over seedlings and securing edges [25]. When the first snowfall is late, some nursery managers rent or use their own snowmaking systems to start making artificial snow at an optimum temperature of −8 °C (yield: 45 m$^3$/h). The protection of seedlings by snow is done in two steps. The first step is to cover the roots of the seedlings by 5 cm to protect most of the seedlings, while the second step is to cover the apical buds of the seedlings by at least 5 cm [25,26]. In the Juniper Forest Nursery in the Province of New Brunswick, Canada, the use of the snowmaking system has become common practice to protect seedlings from winter frost since the early 1990s [70]. With two snowmaking systems, the nursery snows 3.2 hectares with a snow depth of 30 cm (snow quality 3: 370.0 kg/m$^3$) and almost eliminates seedling losses due to frost. The return on investment of the two snowmaking systems was amortized after 1.6 years [70]. In the province of Québec, forest nurseries use different winter protection techniques, and each forest nursery adopts its protection strategy according to its geographical location, the risk of seedling losses, and economic imperatives [71].

## 5. Conclusions

The results of this study, conducted at an operational scale in a forest nursery, showed that the application of different durations of realistic mild spells, as predicted by climate change scenarios in eastern Canada, at the beginning of winter followed by freezing temperatures had no effect on hardening, bud break, and growth during the recovery period of white and black spruce seedlings under optimal growth conditions. On the other hand, with a significant decrease in snow cover towards the end of winter (mid-March), only the seedlings of the two species subjected to the freezing temperature of −20 °C were negatively affected, regardless of the warming treatment. Conversely, the other seedlings of the same treatments subjected to the temperatures 4 °C, −4 °C, and −12 °C showed normal growth (bud burst of all buds on apical and lateral branches, root growth, etc.) and no apparent damage. The mild spells applied also did not lead to a very significant decrease, for example, in leaf nitrogen concentrations and content. Different cultural practices and

protection strategies are proposed to improve frost tolerance and reduce the winter loss of seedlings.

**Supplementary Materials:** The following supporting information can be downloaded at: https://www.mdpi.com/article/10.3390/f13121975/s1. Table S1: Comparisons of adjusted means of growth variables of two-year-old white spruce (WS 2 + 0) and one-year-old black spruce (BS, 1 + 0) seedlings between the sampling date at the beginning (mid-January 2010) and end of winter (mid-March 2010) in response to different freezing temperatures; Table S2: Observed probabilities (*p*-values) and degrees of freedom of the fixed effects related to the different treatments applied before freezing on mineral nutrient concentrations (%) and contents (mg) of different parts (shoot, root, and whole seedling) of two-year-old white spruce (WS 2 + 0) and one-year-old black spruce (BS 1 + 0); Table S3: Comparison of adjusted means between different treatments before freezing related to mineral nutrient concentrations (%) and contents (mg) of different parts (shoot, root, and whole seedling) of two-year-old white spruce (WS, 2 + 0) and one-year-old black spruce (BS, 1 + 0).

**Author Contributions:** Conceptualization, M.S.L. and M.R.; methodology, M.S.L. and M.R.; supervision, M.R. and M.S.L.; project administration, M.R. and M.S.L.; statistical analyses, M.-C.L. and M.S.L.; data curation, M.S.L., M.R. and M.-C.L.; writing—original draft preparation, M.S.L.; writing—review and editing, M.S.L., M.R. and M.-C.L. All authors have read and agreed to the published version of the manuscript.

**Funding:** Funding for this research project was provided by the Direction de la recherche forestière (DRF) of the Ministry of Forests, Wildlife and Parks of the Government of Québec as part of project no. 112310038: Determination of the frost tolerance thresholds of seedlings in winter in relation to climatic extremes and development of a model for predicting the evolution of the hardening state of seedlings in forest nurseries (holder: Mohammed S. Lamhamedi).

**Data Availability Statement:** Not applicable.

**Acknowledgments:** We sincerely thank Denis Gélinas, Pierre Comtois, and all the staff of the Normandin and Grandes-Piles forest nurseries for their exceptional technical assistance during all phases of this project. We thank all the staff of the organic and inorganic chemistry laboratory of the Direction de la recherche forestière (DRF) for the mineral analyses of the samples and the enriching discussions with the chemical managers (Carol De Blois and Denis Langlois). We thank Steeve Pepin for his editorial comments and Charles Gilles Langlois for his continued support of this project.

**Conflicts of Interest:** The authors declare no conflict of interest. The funders had no role in the design of the study; in the collection, analyses, or interpretation of data; in the writing of the manuscript, or in the decision to publish the results.

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
