# Peer review of "Simulation of Episodic Winter Warming on Dehardening of Boreal Forest Seedlings in Northern Forest Nurseries"

_forests, doi:10.3390/f13121975_

Round 1

Author Response

My responses to Reviewer 1's comments are incorporated in the attached file.

Reviewer 2 Report

The authors of this exceptionally long manuscript use a large number of references to support their arguments. Professionally mastered research methodology, as well as data analysis, are the major assets of this manuscript. I only have a few minor comments.

L104-L108: Picea mariana and Picea glauca are introduced. Later, the authors talk about black spruce and white spruce. Although the reader can easily guess which species is which, there is no direct explicit connection - I recommend stating it explicitly.

L221-L225: I recommend emphasizing it more throughout the manuscript.

L340-L343: Full (and translated) name of the Laboratory and Quebec Forest Research Branch is provided in the manuscript twice (also L136-L138). For the second occurrence, I would have listed only the original (or only the translated name).

L350-L351: The excellent data analysis is described in great detail, but (like the entire manuscript) is sometimes not reader-friendly. E.g. I would state separately that "relative electrical conductivity, index of injury and different growth variables at the end of the fall were dependent variables. And forest species, different temperatures (…) and their interaction were explanatory variables/fixed effects…

In general, I recommend shortening and simplifying sentences in the entire manuscript. If you manage to occasionally drop words or whole sentences during this, it will only be to the benefit of the manuscript.

L631: …temperatures. (4°C,…) – remove the period.

Author Response

My responses to Reviewer 2's comments are incorporated in the attached file.
